# The noncoding small RNA SsrA is released by *Vibrio fischeri* and modulates critical host responses

Silvia Moriano-Gutierrez[ID][1,2], Clotilde Bongrand[1], Tara Essock-Burns[ID][1], Leo Wu[1], Margaret J. McFall-Ngai[ID][1], Edward G. Ruby[ID][1]*

1 Pacific Biosciences Research Center, University of Hawai'i at Mānoa, Honolulu Hawai'i, United States of America, 2 Molecular Biosciences and Bioengineering,. University of Hawai'i at Mānoa, Honolulu, Hawai'i, United States of America

* eruby@hawaii.edu

**Data Availability Statement:** All RNA-seq files are available from the NCBI SRA database: accession numbers PRJNA629992 and PRJNA629425.

## Abstract

The regulatory noncoding small RNAs (sRNAs) of bacteria are key elements influencing gene expression; however, there has been little evidence that beneficial bacteria use these molecules to communicate with their animal hosts. We report here that the bacterial sRNA SsrA plays an essential role in the light-organ symbiosis between *Vibrio fischeri* and the squid *Euprymna scolopes*. The symbionts load SsrA into outer membrane vesicles, which are transported specifically into the epithelial cells surrounding the symbiont population in the light organ. Although an SsrA-deletion mutant (Δ*ssrA*) colonized the host to a normal level after 24 h, it produced only 2/10 the luminescence per bacterium, and its persistence began to decline by 48 h. The host's response to colonization by the Δ*ssrA* strain was also abnormal: the epithelial cells underwent premature swelling, and host robustness was reduced. Most notably, when colonized by the Δ*ssrA* strain, the light organ differentially up-regulated 10 genes, including several encoding heightened immune-function or antimicrobial activities. This study reveals the potential for a bacterial symbiont's sRNAs not only to control its own activities but also to trigger critical responses promoting homeostasis in its host. In the absence of this communication, there are dramatic fitness consequences for both partners.

## Introduction

Intracellular bacterial pathogens can activate vertebrate innate immune responses [1–3] by releasing RNA into the cytoplasm. This RNA is sensed by pattern-recognition receptors (PRRs) [4] that trigger the type I interferon (IFN) pathway [5] through retinoic-acid inducible gene-I (RIG-I) signaling [3,6–8]. Certain invertebrates display hallmarks of these vertebrate responses [9]; for example, mollusks, such as oysters or squid, encode homologs of PRRs that sense nucleic acids, as well as other elements involved in signaling-cascade pathways, including RIG-I-like receptors [10–13]. Expression of these genes is up-regulated in response to infection, indicating that the mollusk antimicrobial signaling pathway is complex and reactive

**Funding:** The work was supported by the National Institutes of Health (USA), grants R37AI50661 (MM-N and EGR), R01OD11024 (EGR and MM-N), R01GM135254 (EGR and MM-N), and P20GM125508 (MM-N and EGR). Additional support was provided by the National Science Foundation (USA), grants MCB1608744 (MM-N and EGR) and DBI1828262 (EGR and MMN). The funders had no role in study design, data collection and analysis, decision to publish, or preparation of the manuscript.

**Competing interests:** The authors declare that no competing interests exist.

**Abbreviations:** APO, aposymbiotic; C3, complement protein 3; DMC, Dunn's multiple comparison test; HCR-FISH, fluorescence in situ hybridization chain reaction; IFN, interferon; IRF, interferon-regulatory factor; MAMP, microbe-associated molecular pattern; NF-κB, nuclear factor kappa-light-chain-enhancer of activated B cells; ncRNA, noncoding RNA; OMV, outer membrane vesicle; PGN, peptidoglycan; PRR, pattern-recognition receptor; qRT-PCR, quantitative real-time PCR; RIG-I, retinoic-acid inducible gene-I; RLU, relative light units; RNA-seq, RNA sequencing; sRNA, small RNA; TEM, transmission electron microscopy; TMC, Tukey's multiple comparison test; WT, wild type.

[14,15]. Extracellular pathogens can also induce these pathways by releasing RNA [16] that is protected within outer membrane vesicles (OMVs) [17–24]. This RNA is mainly noncoding (ncRNA), like the regulatory small RNAs (sRNAs) [25] transferred into host cytoplasm by OMVs of *Mycobacterium tuberculosis* [26].

Compared to these pathogenic interactions, there are few reports describing ncRNA signaling mechanisms between beneficial microbes and their hosts [27,28]. Although it is well established that an animal's microbiome is critical to its health, little is known about whether these beneficial bacteria use ncRNA communication to initiate their symbioses, possibly because animal microbiomes typically are phylogenetically complex and difficult to visualize. In contrast, the monospecific light-organ mutualism between the Hawaiian bobtail squid, *Euprymna scolopes*, and the bioluminescent γ-proteobacterium *Vibrio fischeri* offers an experimentally accessible model system for discovering how ncRNAs produced by a beneficial symbiont may be sensed by the host and modulate its responses.

This symbiosis begins when a newly hatched juvenile squid is colonized by planktonic *V. fischeri* cells that enter pores on the surface of the nascent light organ and proceed down a migration path ending at epithelium-lined crypt spaces (Fig 1A). Once there, the bacteria proliferate [29,30] and induce bioluminescence critical to the squid's nocturnal behavior [31]. The initiation of this highly specific association involves a choreographed exchange of signals [32] that changes gene expression in both partners [33,34]. As a result, colonization by *V. fischeri* down-regulates several host antimicrobial responses, including phagocytosis [35], and the production of nitric oxide [36] and halide peroxidase [37]. Nevertheless, the pathways by which these immune adaptations are achieved, and their importance to symbiotic homeostasis, have remained unexplained.

*V. fischeri* OMVs trigger host responses during symbiosis [38], and here we (1) report that these OMVs carry a ncRNA encoded by *ssrA* called tmRNA (SsrA) and (2) visualize this SsrA within the epithelial cells lining the crypts. SsrA is a small stable RNA molecule that, together with its essential chaperon, SmpB [39], participates in the ribosome-rescue system of many bacteria [40,41]. By comparing the host's responses to colonization by either the wild-type (WT) strain or its Δ*ssrA* derivative, we determined that the absence of SsrA sensing within host cells has dramatic negative consequences for the partnership. *V. fischeri* cells that produce OMVs lacking SsrA do not persist in the light organ, and, in the absence of SsrA, the colonization leads to a heightened immune response and a loss of host robustness. Taken together, these data demonstrate the potential for sRNA molecules to be key elements in the language of beneficial host-microbe associations.

## Results and discussion

### The bacterial sRNA SsrA is found within OMVs

In a recent study [33], we reported that symbiotic colonization resulted in transcriptional changes not only in the light organ but also in the expression of host genes in anatomically remote organs. To begin to understand the mechanisms underlying those distal responses, we analyzed the hemolymph of adult squid to detect signal molecules being sent through the body via the circulation. An RNA-sequencing (RNA-seq) study (PRJNA629011) revealed sequences that unexpectedly mapped against the *V. fischeri* genome. This finding indicated that the hemolymph of symbiotic squid carried RNAs produced by the bacterial population of the light organ. Specifically, 166 *V. fischeri* genes were identified, the majority being tRNAs (74%) and ribosomal RNAs (22%), with a lesser complement of small ncRNAs (4%) (Table 1, Fig 1B, S1 Data). We hypothesized that rRNAs and tRNAs are majors components of the circulating RNA population because their secondary structure, provided them greater stability. These

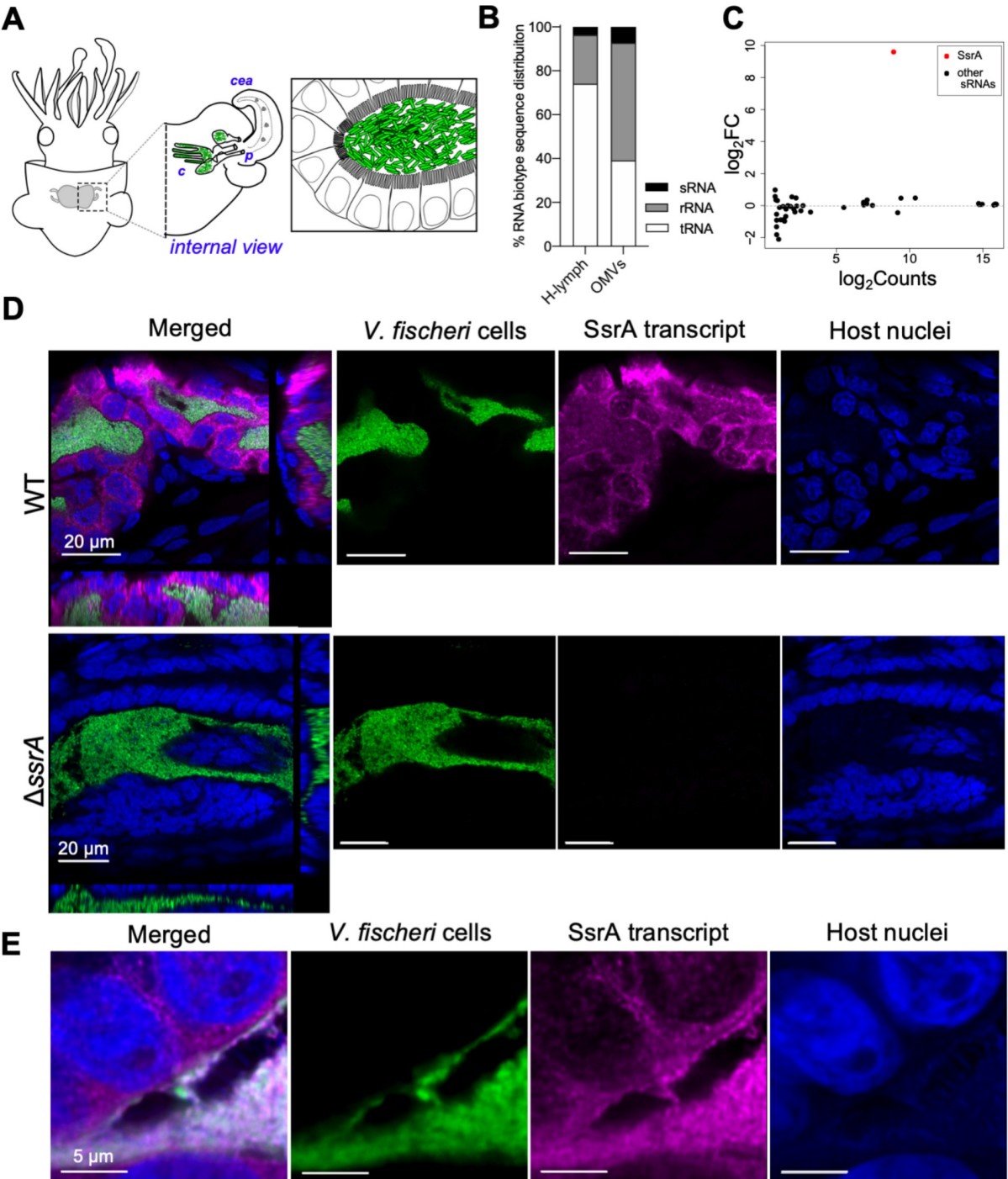

**Fig 1. Symbiont noncoding RNA, SsrA, localizes within the crypt epithelium.** (A) Diagram of a juvenile squid showing the anatomical location (left) and internal aspects (middle) of the light organ, illustrating one of its two pairs of cea, and three entry pores ("p") through which the symbionts reach the migration path to internal crypts ("c"). Gray dots inside the sinus of the cea represent symbiosis-induced trafficking of hemocytes. (Right) Illustration of the close contact between the *V. fischeri* population (green) and the light-organ epithelial cells in a crypt. (B) Relative proportions of types of *V. fischeri* RNAs present in squid H-lymph or in the RNA cargo of OMVs (S1 Data). (C) Volcano-plot representation of fold-change in gene expression (log$_2$FC) of the RNA cargo in OMVs produced by WT or the *ssrA*-deletion mutant *ΔssrA* strain; the only significant difference in RNA content is the presence (in WT) or absence (in *ΔssrA*) of SsrA. Transcripts with evidence for significant differential expression (FDR < 0.05) are colored in red (S1 Data). (D) Localization of symbiont SsrA transcript by confocal microscopy, 24 h after colonization by WT or *ΔssrA* bacteria. Left: merged images with orthogonal views; other panels: images of individual labels. (E) Higher magnification of WT *V. fischeri* cells (green) colonizing the light organ, showing the location of SsrA transcript

(magenta) within the cytoplasm of host epithelial cells. cea, ciliated epithelial appendages; FDR, false discovery rate; H-lymph, hemolymph; OMV, outer membrane vesicle; sRNA, small RNA; WT, wild type.

latter small ncRNAs (sRNAs) were of particular interest to us because this class contains important regulators of gene expression and other cellular activities [42]. Among these, the bacterial translation quality-control molecule, SsrA, was the most abundant in the hemolymph (Table 1).

When OMVs were isolated from a culture of the light-organ symbiont *V. fischeri* strain ES114 [43], ncRNAs representing 73 genomic regions were identified in their contents by Illumina sequencing. The majority of these reads also mapped to ribosomal RNA and tRNA genes (Fig 1B). The remaining RNAs in OMVs were sRNAs (Fig 1B), which were found to have full coverage and, as such, appear not to be degraded. As with the hemolymph samples, SsrA was one of the major species in both *V. fischeri* cells and their OMVs (Table 1) regardless of the growth medium (S1A and S1A' Fig). Furthermore, SsrA was found to be present roughly in the same proportions between the bacterial cell compartment (S1A Fig) and their purified OMVs (S1A' Fig), suggesting there is no significant selective packaging of SsrA within OMVs.

In short, *V. fischeri* symbionts continuously release OMVs containing SsrA into the crypt environment [38,44], from which they may find their way into the circulation, as reported as well in mice [45]. To determine whether this release plays a regulatory role in the light organ, we constructed a *V. fischeri* clean-deletion mutant of *ssrA* (Δ*ssrA*), whose OMVs differed only in the absence of SsrA (Fig 1C, and S1B Fig); similarly, the major species of proteins [46] in OMVs from the two strains were indistinguishable (S1C Fig). Nevertheless, we recognize there may still be differences in low-abundance proteins [46] or in sRNAs that the library preparation could not efficiently record. When compared to its WT parent, the *V. fischeri* Δ*ssrA* mutant had no growth deficiency in either rich or minimal media (S2A Fig), had similar rates of motility (S2B Fig) and respiration (S2C Fig), and initiated colonization normally, but failed to persist as well as WT (Fig 2A). Even though the Δ*ssrA* mutant is outcompeted by its WT parent during co-colonization experiments (S2D Fig), the mutant's colonization ability is unlikely to be affected by its general physiology, probably because this species encodes backup ribosome-rescue systems, ArfAB [47], which allow the normal growth of both the Δ*ssrA* mutant and a mutant in SsrA's specific chaperon, SmpB, (S2A Fig); nevertheless, the absence of SsrA appeared to compromise symbiotic persistence and homeostasis.

## Symbiont SsrA localizes within the crypt epithelium of the light organ

To better understand whether SsrA enters host tissues, we contrasted its absence in the Δ*ssrA* mutant (S2E Fig) to its localization in light organs colonized by either the WT parent or the

**Table 1. List of abundant small, noncoding RNAs .**

| Gene product | Locus tag | No. of reads per fraction[*] | |
|---|---|---|---|
| | | OMV RNA | Hemolymph RNA |
| CsrB1 | VF_2593 | 5,455 (19%) | nd |
| CsrB2 | VF_2577 | 10,804 (37%) | 11 (3%) |
| RnpB | VF_2654 | 3,411 (12%) | 18 (4%) |
| SsrA | VF_2639 | 8,390 (29%) | 306 (70%) |
| SsrS | VF_2651 | 1,234 (4%) | 70 (16%) |
| Ffs | VF_2599 | nd | 31 (7%) |

[*]Percentage of the total number of sRNA reads.

Abbreviations: OMV, outer membrane vesicle; nd, not detected; sRNA, small RNA

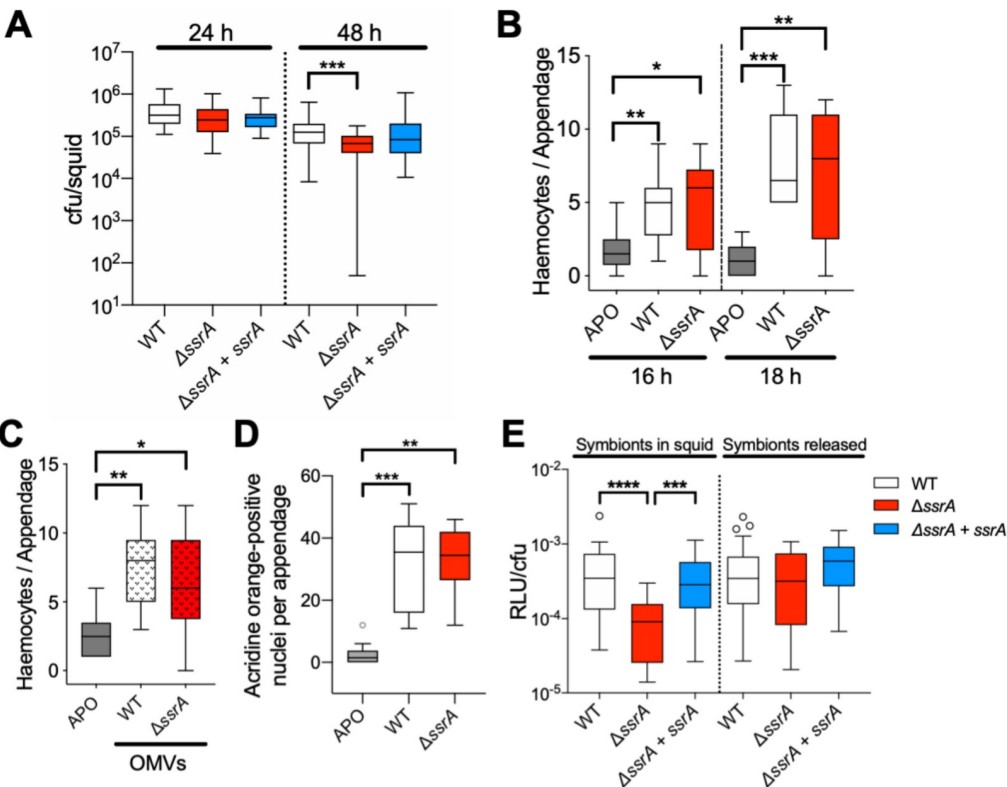

**Fig 2. The Δ*ssrA* mutant is able to initiate colonization normally, but persists poorly.** (A) The number of *V. fischeri* cfu per light organ at 24 or 48 h post colonization in animals colonized by WT, Δ*ssrA*, or the genetically complemented Δ*ssrA* + *ssrA* strains. A 1-way Kruskal–Wallis ANOVA, followed by DMC was performed. Ten squid/condition from six different clutches were used in this experiment ($n = 60$). (B) Levels of hemocyte trafficking into the light organ's anterior appendages at 16 and 18 h post colonization. Significant differences, as indicated by a 1-way ANOVA with TMC ($n = 10$) are shown. (C) Levels of hemocyte trafficking 3 h after exposure to 100 μg of OMVs per ml. Significant differences, as indicated by a 1-way ANOVA with TMC ($n = 10$) are given. (D) The degree of apoptosis in the light organ's ciliated epithelium, as indicated by the number of acridine orange-staining nuclei, in animals that were uncolonized (APO) or colonized by either WT or the Δ*ssrA* strain (S5 Fig). Statistical significance was determined by a 1-way ANOVA, followed by DMC ($n = 10$). (E) Specific luminescence (RLU per cfu) of symbionts either within the light organ, or within a homogenate of the light organ, of a 24-h juvenile. Animals were uncolonized (APO) or colonized by either WT or the Δ*ssrA* strain. Significant differences are given, as indicated by a 1-way ANOVA, followed by DMC. The experiment was repeated twice with the same outcome. *P* value code: ****$<0.0001$; ***$<0.0002$; **$<0.001$; *$<0.021$ for all graphs. Numerical values for all graphs can be found at S2 Data. APO, aposymbiotic; cfu, colony-forming units; DMC, Dunn's multiple comparison test; OMV, outer membrane vesicle; RLU, relative light units; TMC, Tukey's multiple comparison test; WT, wild type.

genetically complemented mutant (Δ*ssrA* + *ssrA*). Using fluorescence in situ hybridization chain reaction (HCR-FISH), SsrA transcripts were found in WT-colonized crypts (Fig 1D; upper panels) whereas no signal was detected in Δ*ssrA*-colonized ones (Fig 1D, lower panels). Significantly, SsrA was observed not only inside the symbiont cells but also within the epithelial cell layer that directly contacted the symbionts. After the majority of the symbionts were vented from the crypts each morning [32], the signal disappeared within minutes (S3 Fig), suggesting that the transcript must be continuously delivered to maintain its level within host cells. Given this apparently rapid turnover of SsrA within the crypt epithelium and the absence of evidence of bacterial lysis within the light-organ crypts by either transmission electron microscopy (TEM) [48] or live-dead stain [49], we hypothesized that the major source of the SsrA found within host cells is OMV-delivered. A higher-magnification image (Fig 1E) revealed abundant SsrA within the cytoplasm (but little detected in the nucleus) of crypt

epithelial cells. In addition, other ncRNAs found within OMVs, such as 16S rRNA, were also observed within the host epithelium (S4 Fig). Significantly, while the symbionts traverse a long epithelium-lined migration path on their way to the crypts (Fig 1A), these host cells show a high degree of localized functional differentiation [49], with only the epithelium lining the crypt becoming labeled with SsrA (S4 Fig). We conclude that the crypt epithelium may be particularly susceptible to bacterial products presented by symbiont-derived OMVs and secreted molecules [50].

## The Δ*ssrA* mutant initiates symbiosis normally and can trigger typical host responses

We next asked whether host cells exhibited any SsrA-dependent responses during the initiation of symbiosis. For instance, colonization by *V. fischeri* cells typically causes symbiont-induced morphogenesis in the light organ [32], including (1) trafficking of macrophage-like hemocytes into the blood sinus of the ciliated epithelial appendages (Fig 1A, middle), induced principally by the peptidoglycan (PGN) monomer [51], and (2) apoptosis in these appendages, triggered primarily by the lipid A portion of lipopolysaccharide [52]. The presence of both these microbe-associated molecular patterns (MAMPs) works synergistically on the two events, which in nature result from colonization or by exposure to *V. fischeri* OMVs [38]. When we compared a colonization by the Δ*ssrA* mutant and its WT parent, or a 3-h exposure to OMVs isolated from those two strains, we observed no difference in either hemocyte trafficking (Fig 2B,C) or apoptosis (Fig 2D and S5 Fig). Thus, the host response to these two MAMPs is not different in the Δ*ssrA* mutation.

## The Δ*ssrA* mutant initiates several abnormal symbiotic responses

WT- and Δ*ssrA*-colonized light organs contained the same number of symbionts by 24 h (Fig 2A); however, Δ*ssrA*-colonized animals emitted only 20% of the WT level of luminescence (Fig 2E). Nevertheless, when each symbiont population was released from its light organ, the light emission produced per bacterium was comparable. This result indicated that, although the Δ*ssrA* symbionts have the same luminescence potential as WT, when they are within the organ their light emission is constrained, possibly owing to oxygen limitation [53].

Another symbiosis-triggered host response is an increase in the volume of the crypt epithelial cells induced by the symbiont's luminescence [54]. To determine whether symbionts lacking SsrA still induced this cell swelling, we compared the cytoplasmic cross-sectional area of epithelial cells in light organs colonized by the WT, Δ*ssrA*, or, as a negative control, a nonluminescent Δ*lux* strain. Unlike the Δ*lux*, at 48 h post colonization both WT- and Δ*ssrA*-colonized squid exhibited normal crypt-cell swelling relative to aposymbiotic (APO) animals (Fig 3A and 3A'). In contrast, after 24 h, only light organs colonized by Δ*ssrA* cells had an increased cytoplasmic area, showing that colonization by a symbiont that produces no SsrA induced a significantly earlier swelling of the crypt epithelium, suggesting that the absence of SsrA sensing generates a dysregulated host response.

Symbiont-induced changes in light-organ gene expression occur within a few hours of colonization [33,55]. To investigate whether this transcriptional response is influenced by the presence of SsrA, we performed a comparative RNA-seq analysis on WT- and Δ*ssrA*-colonized light organs 24 h after colonization. Compared to WT-colonized animals, 10 host genes were significantly up-regulated in the Δ*ssrA*-colonized organs, including typical microbe-responsive genes with known immune-function or antimicrobial activities. These genes encoded laccase-3, several chitinases, and a galaxin-like protein, among others (Fig 3B, S3 Data), and their

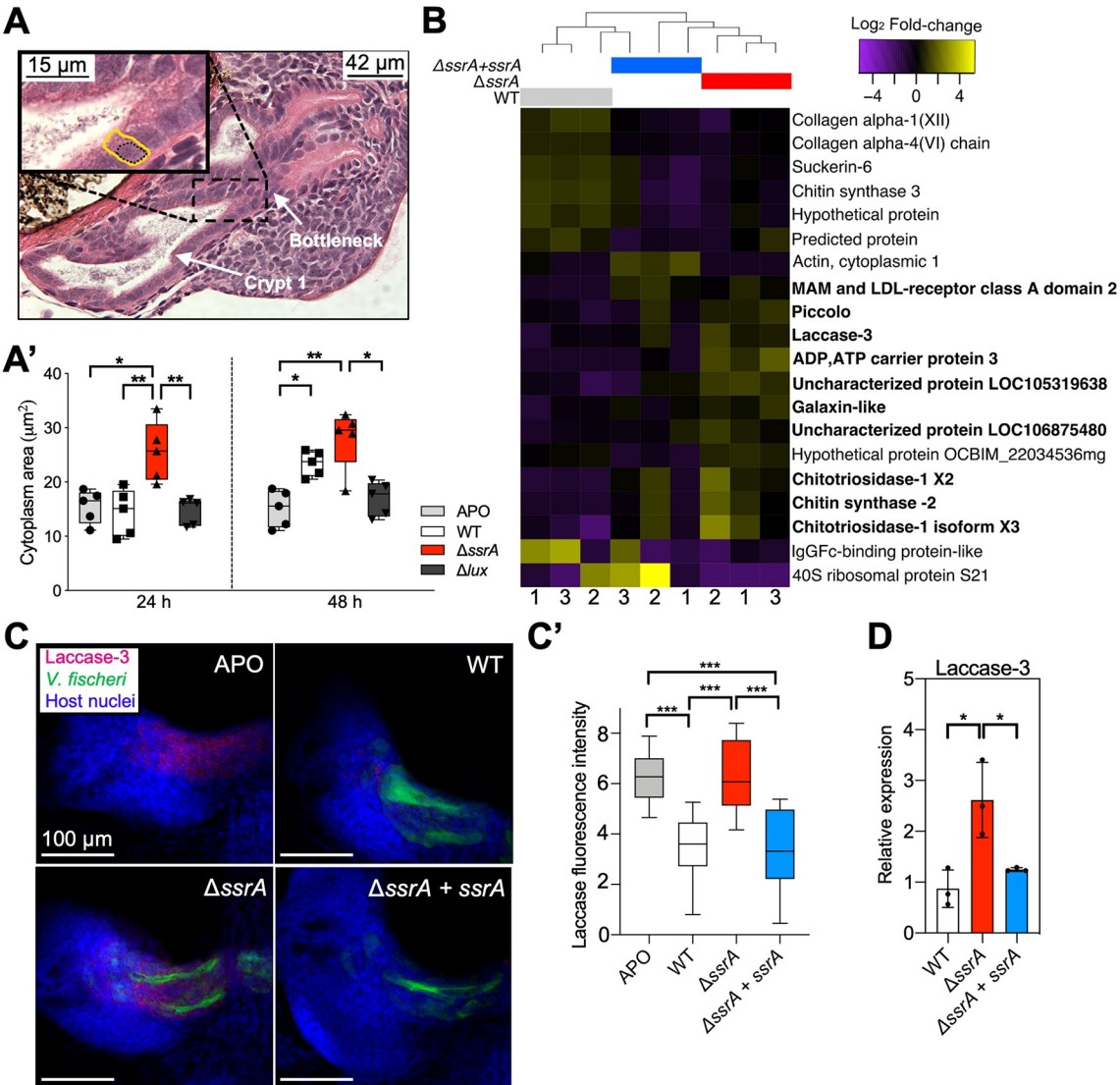

**Fig 3. Host responses to colonization by WT or Δ*ssrA* differ.** (A) Paraffin-section image of a WT-colonized light organ after 48 h, illustrating how crypt-cell cytoplasmic volume was measured. The nuclear area (black dotted line) was subtracted from the total cell area (yellow line). The areas of 10 epithelial cells in crypt 1, just inside of (i.e., distal to) the bottleneck, were measured per light organ. (A') Cytoplasmic volume of the crypt epithelium at 24 and 48 h post inoculation with WT, Δ*ssrA* or Δ*lux* strains, or left uncolonized (APO). (*n* = 5). (B) Heat map depicting fold-change differences in significantly differently expressed genes in light organs colonized by WT, the Δ*ssrA* mutant, or its genetically complemented (Δ*ssrA* + *ssrA*) strain. Genes that are up-regulated in Δ*ssrA*-colonized animals compared to WT-colonized are indicated in bold. The replicate number for each condition (S3 Data) is indicated beneath the heat map. (C) Localization of the laccase-3 transcript (magenta) on one side of the light organ using hybridization chain-reaction fluorescence *in situ* hybridization labeling. Light organs were colonized by the indicated strain of GFP-labeled symbionts (green). (C') Quantification of laccase-3 signal using relative fluorescence intensity of a Z-series image of the light organ (*n* = 9). *P* values were calculated using a 1-way ANOVA with TMC. (D) Relative expression of laccase-3 after 24 h post colonization in light organs colonized by WT, Δ*ssrA*, or Δ*ssrA* + *ssrA*, determined by qRT-PCR. Expression was normalized to ribosomal protein S19 and expressed as 2^ΔΔCT normalized to WT expression. Significant differences are indicated by a 1-way ANOVA with TMC (*n* = 3). Data are presented as the mean ± SD. *P* value code: ****<0.0001; ***<0.0002; **<0.001; *<0.021. Numerical data can be found at S3 Data. APO, aposymbiotic; GFP, green fluorescent protein; qRT-PCR, quantitative real-time PCR; TMC, Tukey's multiple comparison test; WT, wild type.

induction is not inconsistent with the idea that the host treats the Δ*ssrA* colonization as an undesired infection.

Because laccase-3 encodes an extracellular enzyme involved in the synthesis of melanin, a key component of the invertebrate immune response to pathogens [56], we asked where this transcript occurred within the light organ. At 24 h post colonization, the laccase-3 transcript was localized to the crypt epithelium (Fig 3C) in direct contact with the symbionts (Fig 1A). Whereas the HCR signal for laccase-3 was down-regulated after colonization by an SsrA-producing strain (Fig 3C and 3C'), the epithelium's laccase-3 expression remained high if the symbiont failed to produce SsrA. These findings were validated by quantitative real-time PCR (qRT-PCR) (Fig 3D). Thus, delivery of SsrA into the crypt epithelium appears to be required to down-regulate the expression of this, and possibly other, immune defenses. The failure of the Δ*ssrA* symbionts to reduce laccase-3 expression may be tied to their reduced luminescence within the light organ (Fig 2E). We hypothesize that the increased oxidase activity [57] resulting from the higher laccase-3 concentration within Δ*ssrA*-colonized crypts depletes tissue oxygen that otherwise would be available for symbiont light production [53].

## The absence of SsrA sensing, but not SsrA activity in the symbiont, weakens the host

In the bacterial cell, SsrA requires its chaperone, SmpB, to function as part of the stalled-ribosome rescue system [39]. We used this dependency to ask whether the function of SsrA within the symbiont is necessary to induce the SsrA-dependent host responses, by constructing a clean-deletion mutant of *smpB*. Like Δ*ssrA* cells, the Δ*smpB* mutant had no growth defect in culture (S2A Fig), but it expressed normal levels of SsrA (S2E Fig) that accessed the cytoplasm of crypt epithelia (Fig 4A and S6A Fig) similarly to WT (Fig 1D and 1E and S4 Fig). Additionally, the Δ*ssrA*-colonized animal's survival defect can be rescued by co-colonization with WT (S6C Fig). Thus, colonization with the Δ*smpB* mutant results in SsrA delivery to the host without SsrA function in the symbiont, and any host response to Δ*ssrA* that is not also evoked by Δ*smpB* is likely due to a direct, signal-like activity of SsrA within the host cells.

Although not yet fully understood, the pathway by which the SsrA molecule impacts the host appears to be direct, rather than indirect through its activity within the symbionts. This result, together with the normal responses to the Δ*smpB* mutant (Fig 4B, S6B and S6E Fig, S7A and S7A' Fig), provides strong evidence that a critical part of initiating a stable symbiosis is that the host senses, and responds specifically to, the SsrA entering the cytoplasm of the crypt epithelium.

Because of the premature cell swelling and immune-like transcriptional responses of crypt epithelial cells in contact with Δ*ssrA* symbionts, we sought to determine whether the absence of cytoplasmic SsrA in the epithelium compromised the host's health and, if it did, whether SsrA was acting directly. To address these questions, we performed a survival assay on juvenile hatchlings that were colonized either by WT, Δ*ssrA*, Δ*ssrA* + *ssrA*, or Δ*smpB* strains. Those squid colonized by Δ*ssrA*, but not Δ*smpB*, had a survival defect relative to WT-colonized squid (Fig 4B and S6B Fig), indicating that the absence of SsrA within the crypt epithelium, and not the lack of SsrA activity within symbiont cells, compromised the survival of the host. In a similar experiment, the expression of laccase-3 transcript within the Δ*smpB*-colonized light organ was down-regulated like WT, rather than unregulated, as with Δ*ssrA* (S7A and S7A' Fig). Interestingly, the absence of this down-regulation in the Δ*ssrA*-colonized epithelium was not rescued by the SsrA within externally provided WT OMVs (S7B Fig), indicating that curbing of the expression of this immune-defense enzyme likely requires that SsrA be delivered from the symbiont population within the crypts.

An increased immune response can be expected to impose an energetic cost on the host; e.g., when activated, macrophages use ATP more rapidly [58], and defending against a septic

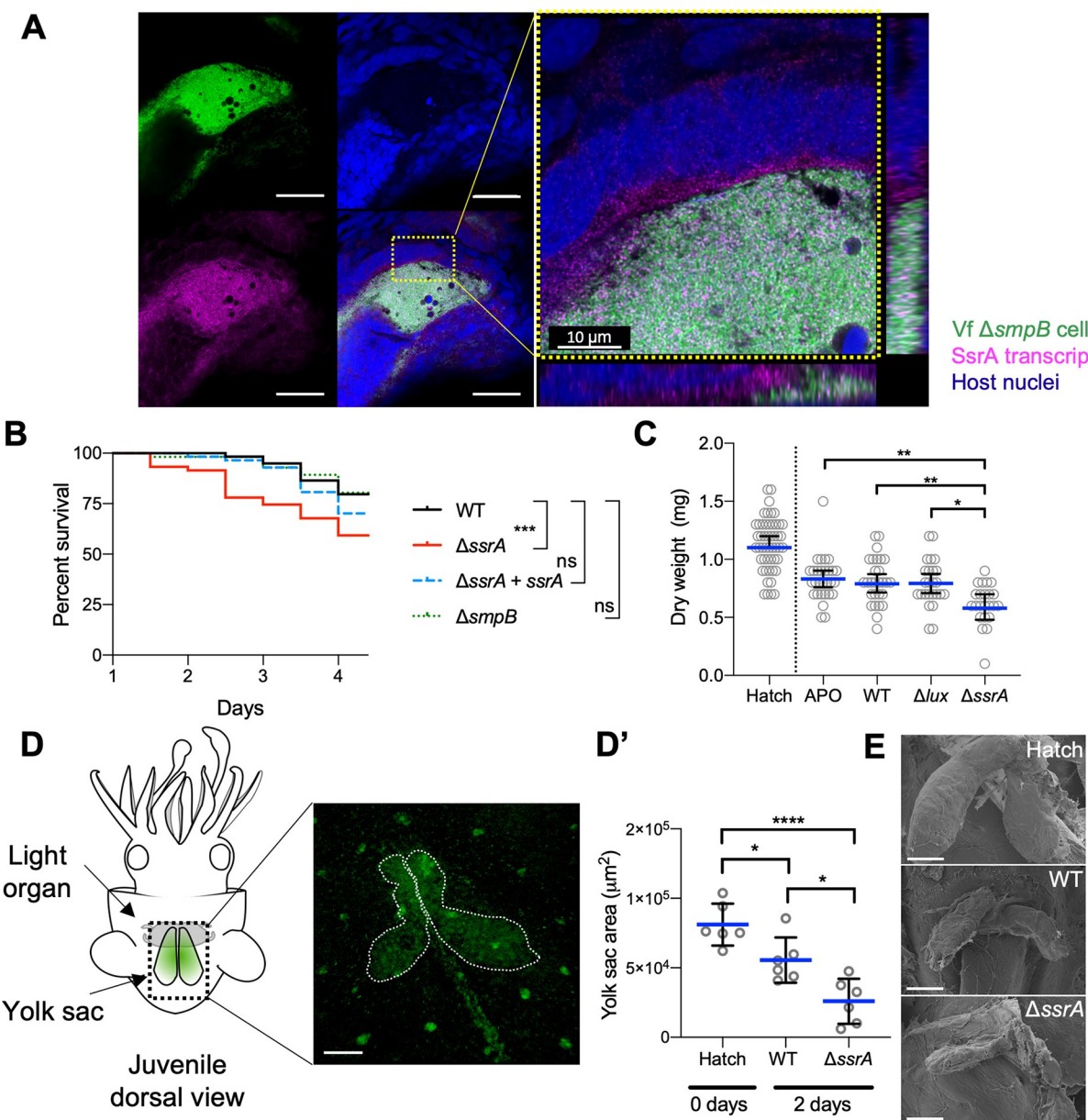

**Fig 4. The absence of SsrA in the epithelium, but not SsrA activity in the symbiont, weakens the host.** (A) Visualization by HCR of SsrA transcript (magenta) in a whole-mount light organ, 24 h after colonization with a GFP-labeled Δ*smpB* strain of Vf (green). A representative confocal image indicates that symbiont SsrA transcript is within the crypt epithelial cells. Scale bar, left panel = 30 μm; see S6A Fig. (B) Kaplan-Meier survival plot of juvenile squid colonized by WT, Δ*ssrA*, the complement (Δ*ssrA + ssrA*), or Δ*smpB* strains. A calculation based on three separate experiments (S6B Fig) is shown, consisting of WT (*n* = 59), Δ*ssrA* (*n* = 59), Δ*ssrA + ssrA* (*n* = 57), or Δ*smpB* (*n* = 56) colonized animals. Survival-curve analysis by a log-rank Mantel-Cox test, with Bonferroni multiple-testing adjustment for pairwise comparisons. *P* value = 0.016. (C) Dry-weight measurements of juvenile squid immediately after hatching ("Hatch") or at 4 d post hatching when kept APO or colonized with WT, Δ*ssrA*, or a dark-mutant (Δ*lux*) strain. Analysis by a 1-way ANOVA with TMC indicated that hatchlings had a significantly greater dry weight compared to all other conditions (*P* < 0.0001). Data are represented as the median, with 95% confidence intervals. (D) Left: dorsal view of a juvenile squid, illustrating the location of the internal yolk sac (dotted box). Right: representative confocal Z-stack image of a hatchling yolk sac stained with the lipophilic lipidspot-488 (green) and depicting how the area (dotted region) was measured; scale bar = 100 μm. (D') Quantification of the internal yolk-sac area was determined from confocal Z-stack images. Data are represented as means ± SD, analyzed by a 1-way ANOVA with TMC. (E) Representative SEM images of the yolk sac of a hatchling squid and animals colonized for 2 d by the WT or the Δ*ssrA*-mutant strain. Scale bar = 100 μm; *P* value code: ****<0.0001; ***<0.0002; **<0.001; *<0.021 for all figures. Numerical values found at S4 Data. APO, aposymbiotic; GFP, green fluorescent protein; HCR, hybridization chain reaction; ns, not significant; SEM, scanning electron microscopy; TMC, Tukey's multiple comparison test; Vf, *V. fischeri*; WT, wild type.

infection increases a tissue's energy demand by 30% [59,60]. Thus, we hypothesized that the effort to reject the ΔssrA colonization entails an energetic cost to the host. As a proxy for such increased metabolic activity, we measured the weight loss of juvenile squid between hatching and 4 d post colonization by either WT, ΔssrA, or Δlux. The latter strain is included as a control because, like ΔssrA, Δlux is unable to maintain a normal colonization level after the first 24 h [61]. We found that animals colonized by the ΔssrA mutant lost weight more rapidly than either APO animals or symbiotic animals colonized by WT or Δlux (Fig 4C). To assure that the differential in weight loss was not due simply to a difference in the activity level of juveniles colonized by the different strains, the respiration rates of the squid were measured. We found that the rate of oxygen consumption was indistinguishable between newly hatched animals or animals colonized for 48 h by WT, Δlux, or ΔssrA (S6D Fig). Thus, neither establishing the symbiosis (i.e., APO versus WT) nor losing the symbiont (i.e., WT versus Δlux) significantly impacted the weight of the juveniles; however, the absence of SsrA (WT versus ΔssrA) did.

In cephalopods, an internal yolk sac provides a reservoir of nutrients that is used by juveniles after hatching, while they learn to hunt prey [62,63]. The animal's observed weight loss over the first 4 d post hatch (Fig 4C) led us to ask whether there was a more rapid depletion of these stored nutrients by ΔssrA-colonized squid. Because of its high lipid content, the size of the yolk sac could be estimated by confocal microscopy using a lipophilic stain (Fig 4D). We found that after 2 d of colonization, in ΔssrA-colonized animals the yolk sac had significantly decreased in area (Fig 4D'), as also confirmed by scanning electron microscopy (Fig 4E). Further, when comparing the yolk sac's size in 2-d-old juveniles, only those colonized by ΔssrA had a significantly smaller yolk sac (S6E Fig), indicating that it is neither the lack of SsrA activity within the symbionts nor the decrease in their number but is instead the failure to deliver SsrA to the host that leads to its faster depletion of yolk-sac resources.

## SsrA may be detected through a host cytosolic RNA sensor

The range of distinct host phenotypes induced by ΔssrA symbionts suggested that, to trigger normal symbiosis development and persistence, the crypt epithelial cells must sense the presence of cytoplasmic SsrA (Fig 1E). Therefore, we asked whether the expression of host cytosolic RNA sensors might be responsible for responding to this SsrA and, in its absence, the lack of this response might contribute to the host's aberrant phenotypes.

Just as how the impact of bacterial sRNA on host cells has been successfully investigated with tissue culture models [64], we chose squid hemocytes as a proxy for the less easily manipulated crypt epithelial cells. Hemocytes are the immune effector cells of mollusks and take up symbiont OMVs both within the crypts (Fig 5A) and in culture [38]. Using isolated hemocytes, we determined the changes in gene expression triggered by OMV-delivered SsrA. Because a change in the levels of complement protein 3 (C3) [65] is a highly conserved innate-immunity reaction, its increased expression was used as a positive control for OMV detection. However, to identify any SsrA-dependent responses, we monitored the expression of the cytosolic RNA sensor RIG-I. As expected, hemocyte expression of C3 was up-regulated after exposure to either WT or ΔssrA OMVs, indicating that both types of vesicles were sensed (Fig 5B); however, only hemocytes that were exposed to WT OMVs responded with a significant increase in RIG-I expression. This differential transcriptional response to SsrA-containing OMVs suggests that their SsrA may quiet the immune response through the RIG-I pathway (Fig 5C).

Because two RIG-I homologs exist in the *E. scolopes* genome [12], the extent and specificity of RNA-sensing mechanisms in this host require further investigation; e.g., functional diversification may have occurred during RIG-I evolution, allowing its paralogs to participate both in communicating with symbionts and/or in antiviral sensing. Additional studies will be required

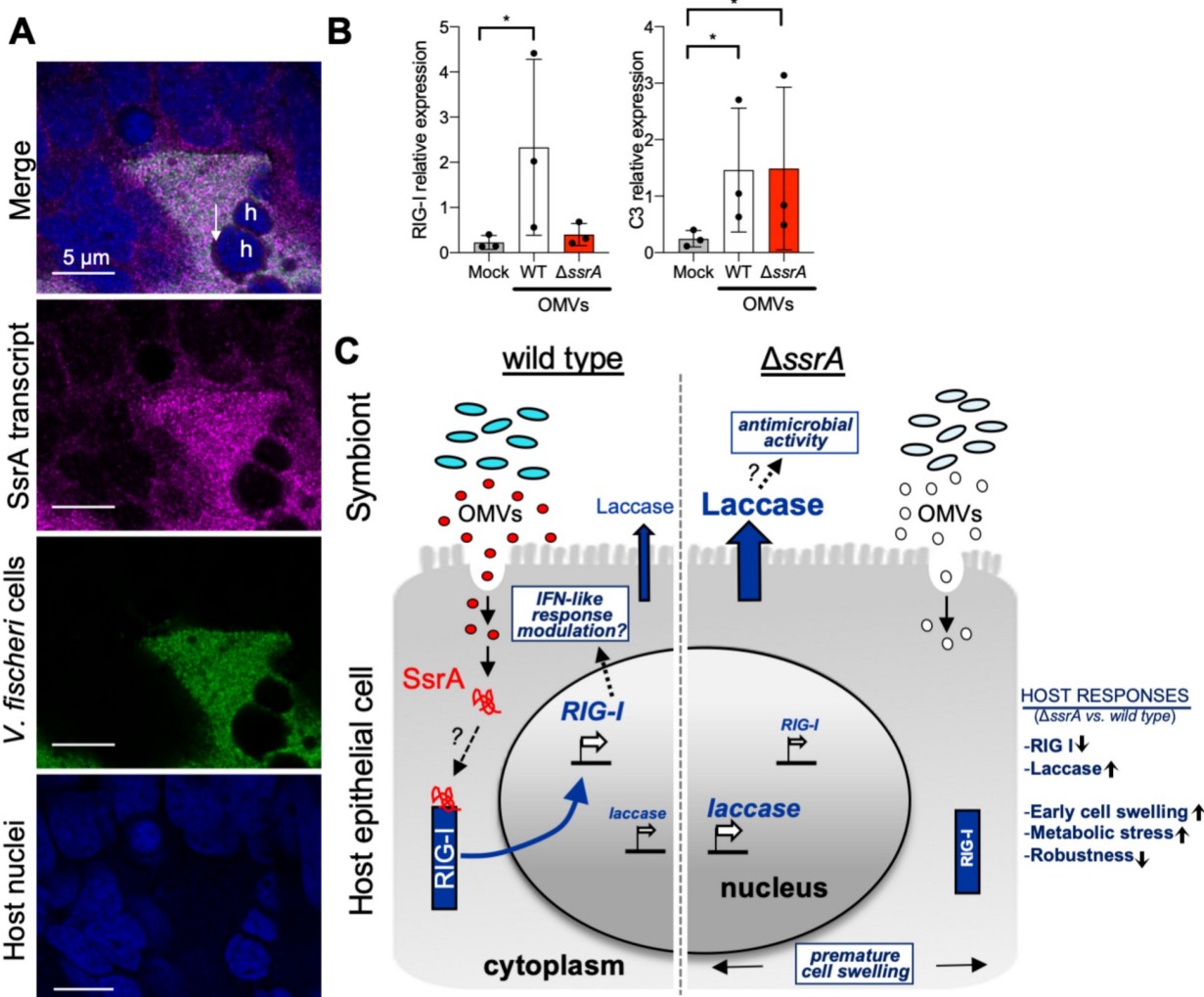

**Fig 5. SsrA taken up by hemocytes may be detected through host cytosolic RNA sensors.** (A) HCR visualization of SsrA transcript (magenta) in a whole-mount light organ, 24 h after colonization with a GFP-labeled WT strain (green). A representative confocal image indicates that symbiont SsrA transcript is within the crypt epithelial cells (nuclei, TO-PRO-3; blue). White arrow indicates symbiont SsrA transcript in a host hemocyte ("h") within the crypt space. (B) Changes in gene expression 30 min after challenging isolated juvenile hemocytes with OMVs purified from exponential cultures of either WT or Δ*ssrA* cells, or after addition of DPBS ("Mock"). Relative expression levels were determined by qPCR for C3 and RIG-I. Error bars = SD, ($n$ = 3); $P$ value code: $^*<0.021$ (S5 Data). (C) A hypothetical model for SsrA modulation of host immune response. During WT colonization, OMVs containing SsrA enter the host cell. The OMV cargo is released into the cytoplasm, where SsrA associates with RIG-I, triggering a signaling cascade that induces RIG-I's own up-regulation as well as the activation of an IFN-like response for symbiont modulation. During Δ*ssrA* colonization, there is no SsrA to associate with RIG-I. As a consequence, there is no modulation of IFN response, leading to inflammation. This result leads to a continued production of antibacterial laccase and cell swelling and an overall diminished robustness of the host due to the rapid depletion of its yolk sac, resulting from the demands of the increased immune response. C3, complement protein 3; DPBS, Dulbecco's phosphate-buffered saline; GFP, green fluorescent protein; HCR, hybridization chain reaction; IFN, interferon; OMV, outer membrane vesicle; qPCR, quantitative PCR; RIG-I, retinoic-acid inducible gene-I; WT, wild type.

to determine the actual in vivo mechanisms of RIG-I-associated signaling, as well as any potentially differential response to SsrA sensing between immune cells, such as hemocytes, and epithelial cells.

The SsrA-dependent induction of the gene encoding RIG-I (Fig 5B) leads us to speculate that (1) a functional RIG-I signaling pathway exists in the Hawaiian bobtail squid and (2), as in other mollusks challenged with viral RNA [66], the up-regulation of RIG-I indicates that this RNA-sensor protein is involved in the recognition of symbiont SsrA. In vertebrates, RIG-I

activates NF-κB (nuclear factor kappa-light-chain-enhancer of activated B cells) and IRF (interferon-regulatory factor) transcription factors, which coordinately regulate the expression of type I IFNs [67]. Although no IFN homologs have been identified in the host, genes encoding several key elements of the IFN pathway are present in the *E. scolopes* genome, including those involved in JAK/STAT signaling and IRFs [12]. For this reason, it has been hypothesized that the functional role of IFN in the squid may be assumed by unannotated genes without a recognizable homology to vertebrate IFN. Thus, we hypothesize that RIG-I may function as a PRR that recognizes symbiont SsrA and acts as a crucial trigger for downstream signaling cascades in the squid (Fig 5C). As a consequence, during Δ*ssrA* colonization, when there is no SsrA associated with RIG-I, a modulation of the host immune responses does not occur. Although they also affect aspects of this response, the Δ*ssrA*-associated differential modulation of RIG-I is apparently not a response to a decreased delivery of symbiont MAMPs (i.e., PGN monomer and lipid A); specifically, colonization by either WT or Δ*ssrA* symbionts induced an equivalent level of both hemocyte trafficking and apoptosis of the ciliated field (Fig 2C and 2D), two symbiosis phenotypes whose induction requires exposure to these MAMPs [68,69]. Furthermore, sensing of bacterial RNA through RIG-I has recently been observed also in other systems [1–3,70,71], indicating that host recognition of beneficial or pathogenic sources of RNAs might be a conserved mechanism by which the host shapes its responses according to not only the identity but also the physiological state of the bacteria encountered.

The mechanism(s) by which SsrA is sensed within the epithelium remains to be determined, but the possibilities include secondary structure or sequence specificity playing a role in the recognition of SsrA by the host. The sequences of SsrAs are relatively conserved across bacteria but, like 16S, can be used to identify different species [72,73]; thus, an investigation of the degree to which host cells may be able to discern SsrA molecules they sense from specific bacteria will be the subject of future studies.

## Conclusions

To successfully initiate a symbiotic association, beneficial bacteria must restrain the immune system of their host; for example, certain plant symbionts use sRNAs to control host responses and foster a cooperative colonization [27,28]. However, no example of beneficial bacteria using RNA to temper an animal's immune response has been reported, except as a result of chimeric genetic engineering [74]. Here, we provide evidence that transmitting an sRNA signal to modulate an animal's defenses is a key element underlying symbiotic homeostasis and persistence. Specifically, we hypothesize that, in a light organ colonized by *V. fischeri*, the RNA sensor RIG-I is activated by SsrA-containing OMVs, avoiding both a dysregulation of normal responses and a heightened immune reaction that would affect the stability of the association (Fig 5C). Whether and how other symbiont RNAs are sensed by the host and lead to specific tissue responses will require further exploration. Nevertheless, we anticipate that host recognition of sRNAs will emerge as a major new category of communication between symbionts and the tissues they inhabit.

## Materials and methods

### Light-organ colonization assays

The breeding colony of Hawaiian bobtail squid (*E. scolopes*) was supplied by collecting adult animals from Maunalua Bay, Oahu, Hawai'i, and transferring them to the Kewalo Marine Laboratory in sun-lite, outdoor, flow-through seawater tanks. Adult females laid egg clutches that were kept in seawater and maintained on a natural 12:12-h light:dark cycle.

Juvenile squid from the breeding colony were collected within minutes of hatching and placed in filter-sterilized ocean water (FSOW). Within 2 h of hatching, juveniles were either made symbiotic (SYM) by overnight exposure to *V. fischeri* strain ES114 [43], WT or derived mutant strains, suspended in FSOW, or kept APO in FSOW without additions. Animals were maintained on a 12:12-h light:dark cycle. To prepare bacterial inocula for colonization, strains were cultured overnight in Luria-Bertani salt medium (LBS) [75] with any appropriate antibiotic selection, if necessary. These cells were subcultured into seawater tryptone medium (SWT) [43] and grown to mid-log phase at 28 ˚C with 220-rpm shaking. The final inoculum was a dilution of this subculture into FSOW to achieve a concentration of 4,000–8,000 cfu/ml. Colonization of the host was monitored by checking for animal luminescence with a TD 20/20 luminometer (Turner Designs, Sunnyvale, CA). Unless otherwise indicated, SYM or APO juvenile animals were analyzed at 24 h post colonization (i.e., 2 h after dusk). The squid were anesthetized in seawater containing 2% ethanol and either flash-frozen and stored at −80 ˚C in RNAlater (Sigma-Aldrich) until further processing, as previously described [76], or fixed overnight in 4% paraformaldehyde (PFA) in marine phosphate-buffered saline (mPBS: 450 mM NaCl, 50 mM sodium phosphate buffer [pH 7.4]).

## Generation of bacterial mutants

The WT *V. fischeri* symbiont strain ES114 [43] was used as the parental strain for all mutant constructions. The enzymes used for cloning were platinum Taq polymerase (Invitrogen), Fast-digest restriction enzymes (Thermo Fischer Scientific), and a T4 DNA ligase (New England Biolabs [NEB]). The SsrA mutant was constructed by allelic exchange with pSMG3, a derivative of the counter-selectable suicide vector pKV363 [77,78]. To build pSMG3, we amplified two fragments: PCRa, approximately 600 bp upstream of SsrA; and PCRb, approximately 500 bp downstream of SsrA (S1 Table). Both products were digested with BamHI at 37 ˚C for 1 h and ligated overnight at 4 ˚C. The ligated product was amplified (PCRab) and inserted between EcoRI and XhoI of pKV363. To construct plasmid pSMG5, which complement the *ssrA* deletion, we amplified a fragment of ES114 gDNA, including the *ssrA* and *smpB* encoding locus. The PCR product was digested with KpnI and XbaI and inserted into those sites in pVSV105 [79]. The SmpB mutant was constructed using a sucrose-based selection with pCBNR36, a derivative of the pSMV3 vector [46]. To build this vector, we amplified two fragments: PCRa, approximately 900 bp upstream of *smpB*; and PCRb, approximately 800 bp downstream of *smpB* (S1 Table). Both products were digested with BamHI at 37 ˚C for 1 h and ligated overnight at 4 ˚C. The ligated product was amplified and inserted between ApaI and SpeI of pSMV3. When necessary, antibiotics were added to media at the following concentrations: 50 and 100 µg ml$^{-1}$ for kanamycin and 25 and 2.5 µg ml$^{-1}$ for chloramphenicol in, respectively, Luria-Bertani broth (LB: 950 ml deionized water, 10 g Bacto-Tryptone, 5 g yeast extract, 10 g NaCl, 50 ml of 1 M Tris [pH 7.5–7.6]) or LBS [75].

To label strains for fluorescence imaging, pVSV102 encoding GFP and a kanamycin-resistance expression cassette was transferred from *E. coli* DH5α to each *V. fischeri* receiver strain by triparental mating [77] using the conjugative helper strain CC118 λ*pir*, as described previously [79].

## Bacterial growth assays

Cells were grown in LBS medium to an OD of 0.6 ± 0.1 and then diluted to an optical density at 600 nm ($OD_{600}$) of 0.02 with either fresh LBS or a minimal medium (MSM: composed of 1 g of Bacto-Tryptone, 20 g of NaCl, and 50 ml of 1 M Tris-HCl buffer [pH 7.5] per liter of deionized water). Under some experimental conditions, LBS was supplemented with glycerol (32.6

mM) or *N*-acetyl-glucosamine (GlcNAc; 10 mM). Growth of 1-ml cultures in 24-well plates was monitored at $OD_{600}$ using a GENiosPro plate reader (Tecan, Morrisville, NC) with continuous shaking at 28 ˚C. Absorbance readings were corrected for a nonstandard path length by linear transformation.

## Host RNA extraction and sequencing

For RNA extraction, 20 juvenile light organs were pooled, and total extracted RNA was purified using QIAGEN RNeasy columns, immediately followed by treatment with TURBO DNase (Thermo Fisher Scientific). The RNA concentration for each sample was then determined with a Qubit RNA BR assay kit (Invitrogen). The Illumina TruSeq Stranded mRNA Sample Prep with polyA selection v4.0 protocol was used for library preparation. Sequencing of light-organ tissue samples was performed at the NYU Genome Center on an Illumina HiSeq 4000 using a paired-end, 100-nucleotide-length run mode.

## Estimation of transcript abundance and differential-expression analysis

Reads from the RNA-seq analyses were mapped against the host reference transcriptome [12] with bowtie2 [80], and their relative expression values were estimated with RSEM software [81]. To identify the differentially expressed transcripts, the R package edgeR [82] was used for the statistical analysis of the RNA-seq data, employing a false discovery rate (FDR) threshold of 0.05. Heat maps of expression values and hierarchical clustering were created with heatmap3 [83] in the R environment.

## HCR-FISH

Fixed juvenile squid were washed three times for 30 min in mPBS prior to dissection of the host tissues. HCR-FISH probes (version3 chemistry [84]) specific for the transcripts encoding either the host's laccase or the *V*. *fischeri* 16S RNA (S1 Table) were designed and provided by Molecular Instruments (www.molecularinstruments.com). Juvenile squid were collected 24 h post colonization under the standard procedures explained above. The light organs were then dissected out and the hybridization procedure was followed as described previously [33]. Samples were counterstained overnight with TO-PRO-3 (Thermo Fisher Scientific) to label host nuclei, and imaged using a Zeiss LSM 710 upright laser-scanning confocal microscope (Carl Zeiss AG, Jena, Germany) located at the University of Hawaiʻi-Mānoa (UHM) Kewalo Marine Laboratory. Fluorescence intensity for all sections of each Z-stack was measured using FIJI [85].

For symbiont SsrA transcript detection, HCR-FISH probes (version2 chemistry [86]) were designed and provided by Molecular Instruments. The hybridization procedure was followed as described [87] and, after counterstaining with TO-PRO-3, the samples were imaged using an upright Leica SP8 confocal microscope (Leica Camera AG, Wetzlar, Germany). Images were adjusted to optimize visual resolution using the Lightning Adaptive deconvolution, and the Leica LasX software, located at UHM.

## qRT-PCR

Gene-expression changes were confirmed by qRT-PCR using LightCycler 480 SYBR Green I Master Mix (Roche) and the same total RNA extracts described previously. Synthesis of the single-stranded complementary DNA was performed with SMART MMLV Reverse Transcriptase (Clontech) using either Oligo(dT)12–18 primers (Invitrogen) for host-gene expression or random hexamers (Invitrogen) for symbiont-gene expression. Following the MIQE

guidelines [88], all reactions were performed with no-RT and no-template controls to confirm that the reaction mixtures were not contaminated. Specific primers (S1 Table) were designed with Primer3plus [89]. The amplification efficiency was determined by in-run standard curves, with a 10-fold dilution template. Each reaction was performed in duplicate with a starting level of 12.5-ng cDNA. The generation of specific PCR products was confirmed by melting-curve analysis. Expression analyses of candidate genes were normalized to either ribosomal protein S19 for host-gene expression analysis, or to polymerase A for symbiont-gene expression analyses. Bar graphs of expression values were produced with GraphPad Prism v8.00 software.

## Paraffin sectioning and histology

Squid were collected after 24 h or 48 h post colonization and fixed in 4% PFA in mPBS, and the light organs were dissected out and dehydrated by serial washes in ethanol. Afterward, the light organs were embedded in paraffin wax, histologically sectioned (5 μm), stained with hematoxylin and eosin, and mounted on slides at the Microscopy and Imaging Core (MICRO) facility of UHM. FIJI [85] was used to measure the cytoplasmic area of light-organ epithelial cells by subtraction of the nucleus area from the total cell area. Five light organs were analyzed, and 10 cells on each border of the crypt side of the bottleneck region of crypt #1 were measured per light organ.

## Dry-weight measurement

Squid were collected at hatching and, at 4 d post colonization, anesthetized in seawater containing 2% ethanol and flash-frozen until further processing. Each squid was placed in a pre-weighted aluminum foil tray, dried at 90°C, and weighed on an Ohaus AX124 balance until a constant dry-weight value had been reached.

## Scanning electron microscopy

Squid were collected 48 h post colonization, fixed and washed in mPBS, before dehydrating through an ethanol series, and critical-point dried as previously described [90]. The samples were mounted on stubs, gold sputter-coated, and viewed with a Hitachi S-4800 FESEM scanning electron microscope at the UHM MICRO facility.

## Yolk-sac staining and measurement

Squid were collected after 48 h of colonization and incubated at room temperature for 2 h in 1:1,000 of the lipid stain, lipidspot488 (Biotium). Afterward, the squid were washed in seawater, anesthetized in seawater containing 2% ethanol, and imaged using a Zeiss LSM 710 confocal microscope. Z-stack images were acquired and the area of the internal yolk sac measured using FIJI [85] from the sum slices of each Z-stack.

## Purification of OMVs

*V. fischeri* cultures were grown at 28 °C in LBS until late exponential phase (OD $\simeq$ 3). The cells were removed by low-speed centrifugation (8,000$g$) at 4 °C, and the culture supernatant was filtered through a 0.22-μm-pore-size PVDF membrane filter (Millipore). OMVs were then collected from the filtered supernatant by centrifugation for 2 h in a TLA-45 rotor using a Max-XP ultracentrifuge (Beckman Coulter) at 180,000$g$ and 4 °C. The pelleted OMVs were washed by resuspension in Dulbecco's phosphate-buffered saline (DPBS) with added salt (0.4 M NaCl) [38] and re-centrifugation at 200,000$g$ for 1 h at 4 °C using either a MLA-50 or TLA-110 rotor in an Optima-XP centrifuge (Beckman Coulter). The resulting pellets were resuspended in

saline DPBS and filter-sterilized through 0.45-μm-pore-size PVDF membrane filter (Millipore) before storing at −80 ˚C. Before OMVs were added to live squid, they were further purified with a sucrose density gradient as previously described [38]. To estimate the OMV concentration, total protein of the sample was determined with the Qubit Protein Assay Kit (Invitrogen).

## Protein gel electrophoresis

Samples containing 10 μg of total protein from *E. scolopes* OMVs were loaded onto a precast 12% bis-tris polyacrylamide gel (Invitrogen), together with a Precision Plus Protein standard (Bio-Rad). The proteins were separated for 1 h at 150 V at 4 ˚C in a Mini-Vertical Electrophoresis System (Bio-Rad), stained overnight in 90% Protoblue Safe (National Diagnostics,) in ethanol, rinsed in deionized water, and imaged with GelDoc-It (UVP) system.

## Sequencing of RNA extracted from OMVs

To determine the nature of their RNA cargo, 500 μL (2,500 μg of protein per ml) of purified OMVs were first treated with 4 mg of RNAseA (Promega) per ml for 10 min at 37 ˚C to remove any surface contamination. The added RNase was then inhibited by the addition of 1 μl of Murine RNase inhibitor (NEB). The RNA within these treated OMVs was purified using a mirVana PARIS kit (Invitrogen), followed by DNAse I treatment (Thermo Fisher Scientific). The RNA concentration for each sample was then determined with a Qubit RNA BR assay kit (Invitrogen). Library preparation and sequencing was performed by SeqMatic (Fremont, CA), with paired-end stranded RNA ($2 \times 75$ bp). Size selection of the library with inserts smaller than 300 nucleotides was performed before sequencing on an Illumina MySeq platform. Reads were mapped to the *V. fischeri* genome (GenBank: CP000020, CP000021, and CP000022), and their relative abundance was estimated with Feature Counts [91]. Differential-expression analysis was performed using the R package edgeR [82] with an FDR threshold of 0.05. Heat maps of expression values were originated with the R package heatmap3 [83].

## Isolation of host hemocytes and purification of their RNA

Hemocytes from APO juveniles were isolated as described previously [92]. For expression analysis, hemocytes from 20 juveniles were pooled and spread into 12-well plates (Millipore-Sigma), allowed to adhere for 20 min at room temperature, and washed three times in Squid-Ringer's solution (SRS: 530 mM NaCl, 10 mM $CaCl_2$, 10 mM KCl, 25 mM $MgCl_2$, and 10 mM HEPES buffer [pH 7.5]). After these washes, purified OMVs from either the WT or Δ*ssrA* strain were delivered at a concentration of 50 μg/ml of SRS and incubated for 30 min at room temperature. For the mock condition, the same volume of saline DPBS was delivered to the hemocytes. Following the incubation, hemocytes were washed in SRS once, and TRIzol Reagent (Invitrogen) was immediately added and incubated for 5 min, and the sample was kept at −80 ˚C until further processing. For RNA extraction, 200 μl of chloroform was added to the sample, which was vortexed, incubated for 5 min, and centrifuged at 120,000*g*. The upper aqueous phase was removed and, after adding 1 vol of 100% ethanol, placed on a silica spin column from the Monarch RNA clean-up kit (NEB) following standard procedures. The RNA was then eluted in 20 μl of nuclease-free water, and the RNA concentration of each sample was determined with a Qubit RNA BR assay kit. Subsequent synthesis of cDNA and qPCR reactions were performed as described above. The *E. scolopes* genome encodes two RIG-I genes; the RIG-I homolog with greater sequence identity to the human RIG-I sequence (O95786-1) was chosen for primer design.

## Hemocyte trafficking assay

Juvenile squid were collected at 16 and 18 h post colonization and fixed as described above. Light organs were dissected out and permeabilized overnight at 4 ˚C in 1% Triton X-100 (Sigma-Aldrich) in mPBS. Hemocytes were stained as previously described [51], with a solution of 1 mg of deoxyribonuclease I conjugated to Alexa Fluor 488 (Thermo Fisher Scientific) per ml of 1% Triton X mPBS for 24 h at 4 ˚C. Samples were counterstained with rhodamine phalloidin (Invitrogen) to visualize the actin cytoskeleton. Hemocytes that had migrated into the blood sinus of the light-organ appendages were visualized and counted using a Zeiss LSM 710 confocal microscope.

## Apoptosis assay

Juvenile squid were collected at 14 h post colonization, anesthetized in 2% ethanol, and placed in 0.0001% acridine orange (Invitrogen, A1301) in seawater for 1 min, as previously described [52]. Light-organ appendages were visualized to determine the number of acridine orange–positive apoptotic nuclei using fluorescence confocal microscopy.

## Squid survival assay

Juvenile squid were colonized following standard procedures. In general, to control for inter-clutch variation, three independent experiments were performed using juveniles from three different clutches. After overnight inoculation with the appropriate strains, squid were transferred into clean glass scintillation vials containing 4 ml of FSOW. Each morning for the duration of the experiment, the squid were transferred into new vials with 4 ml of fresh FSOW but were not fed. Under these conditions, the squid survive until the nutrients in their internal yolk sac are depleted. Approximately every 12 h, colonization of the light organ was monitored by checking for animal luminescence with a TD 20/20 luminometer, and the squid's viability was assessed by recording their responsiveness.

## Measurement of bacterial and host respiration rates

Respiration-rate assays were performed using a digital respirometry system (Model 10, Rank Brothers, Cambridge, United Kingdom), whose data were collected via the analog-digital interface ADC-20 Picolog 1216 data logger (Picolog PicoTechnology, Cambridgeshire, UK). Prior to data collection, the oxygen sensor was calibrated at 100% with air-saturated deionized water. The seawater in the respirometer chamber was fully aerated prior to adding the squid and continuously stirred to maintain a uniform oxygen concentration during the measurement. The linear rate of decline in the oxygen concentration within the sealed chamber was used to calculate oxygen-consumption rates.

For the determination of bacterial respiration rates, overnight LBS cultures were diluted 1:100 in SWT and grown at 28 ˚C until an $OD_{600}$ of 0.3 for replicate #1, 0.4 for replicate #2, or 0.5 for replicate #3. One ml of culture was placed in the chamber, and the rate of decline in the oxygen concentration was measured. The final respiration rate was normalized to the OD as follows: $\Delta O_2(t_0 - t_n)/\Delta OD(t_n - t_0)$, where $t_0$ is the time the measurements started, and $t_n$ is the time the measurements ended. For the squid respiration-rate measurement, animals were placed in the chamber with 1 ml of seawater, and the measurement made without stirring to avoid disturbing the animal.

## Statistical analysis

All data are expressed either as mean and standard deviation or as median with 95% confident intervals. A normality test was applied, where appropriate, to ensure a normal distribution of

the data. A 1-way ANOVA or Kruskal–Wallis ANOVA was used for statistical analysis. The data were considered significant at a *P* value < 0.05. When appropriate, *P* values were adjusted for multiple comparison. The sample number (*n*) indicates the number of independent biological samples tested. Independent experimental replicates are indicated when performed. Information on relevant statistical analysis is provided for each experiment in the figure legends.

### Ethics statement

Adult *E. scolopes* were collected in Oahu (Hawaiʻi) and bred in the laboratory. While the IACUC committee of the University of Hawaiʻi at Mānoa only reviews vertebrate protocols, all squid facilities and experiments reported here have been reviewed by, and conform to the relevant standards established by, the University's senior veterinarian.

### Supporting information

**S1 Fig. The symbiont sRNA SsrA is found in the squid circulatory system and within symbiont OMVs.** (A) qPCR measurements of SsrA expression by WT *V. fischeri* grown in three different media: a tryptone-based medium (LBS) or LBS with the addition of either glycerol (32.6 mM) or GlcNAc (10 mM). Data are presented as the number of transcript copies per cfu in late log phase (*n* = 3). (A') qPCR measurements of SsrA within purified OMVs, presented as the number of transcript copies per volume of purified OMV preparation (*n* = 3). S6 Data. (B) Heat map of expression levels of *V. fischeri* RNA detected in squid hemolymph and in the RNA contents of OMVs. Hemolymph was collected from adult field-caught animals. OMVs were purified from cultures of WT *V. fischeri* or its Δ*ssrA* derivative (S1 Data). (C) Soluble proteins present in purified OMVs isolated from cultures of WT or Δ*ssrA* cells (S1 raw image). cfu, colony-forming units; GlcNAc, *N*-acetyl-glucosamine; LBS, Luria-Bertani salt medium; OMV, outer membrane vesicle; qPCR, quantitative PCR; sRNA, small RNA; WT, wild type. (TIFF)

**S2 Fig. Effects of SsrA deletion on *V. fischeri* cells.** (A) Growth characteristics in (left) the tryptone-based medium LBS or (right) a minimal-salts medium, by the WT *V. fischeri* strain ES114 (WT), the Δ*ssrA* mutant derivative, its genetic complement (Δ*ssrA* + *ssrA*), and a deletion mutant (Δ*smpB*) of the SsrA chaperone, SmpB. (B) Rates of motility in soft agar of WT or Δ*ssrA* cells. The diameter of the outer ring was measured at 3 and 7 h post inoculation. Data are presented as the mean ± SD. (C) Normalized respiration rates of WT, Δ*ssrA*, Δ*ssrA* + *ssrA*, and a nonluminescent, *lux*-deletion mutant (Δ*lux*) in SWT medium. Data are presented as the mean ± SD. (D) RCI between WT and Δ*ssrA* in co-inoculated light organs after 24, 48, and 72 h. The RCI was calculated as the ratio of the two strains in the light organ, divided by their ratio in the inoculum. RCIs that are significantly different from zero are indicated (*t* test: **P value < 0.0071; ***P value < 0.0001). (E) Relative expression of *ssrA* and *smpB* transcripts by cells of WT and its mutant derivatives during the exponential phase of growth (OD$_{600}$ between 0.65 and 0.74) in LBS medium. Expression was normalized to polymerase A and expressed as 2ΔΔCT. Significant differences are indicated by letters, based on a Bonferroni multiple-testing adjustment for pairwise comparisons. *P* value = 0.0083. The genetic complementation Δ*ssrA* + *ssrA*, carries on a plasmid a copy of both *ssrA* and *smpB*. Numerical values found at S6 Data. LBS, Luria-Bertani salt medium; OD600, optical density at 600 nm; RCI, relative competitive index; WT, wild type. (TIFF)

**S3 Fig. Localization of the symbiont's SsrA transcript before and after symbiont expulsion from the light organ.** Representative confocal microscopy images localizing symbiont SsrA

(green) by HCR 30 min before (top) or 30 min after (bottom) symbiont expulsion. Light organs were colonized by WT *V. fischeri*. HCR, hybridization chain reaction; WT, wild type. (TIFF)

**S4 Fig. Localization of the symbiont's 16S and SsrA transcripts within the host's light-organ epithelial cells.** Representative confocal microscopy images with orthogonal projections localizing symbiont SsrA (green) and 16S (magenta) transcripts within the crypt epithelium of light organs colonized by WT, Δ*ssrA*, or Δ*ssrA* + *ssrA*, compared to the HCR hairpin negative control; host nuclei (blue). Scale bars = 20 μm. HCR, hybridization chain reaction; WT, wild type. (TIFF)

**S5 Fig. Induction of apoptosis in the light-organ appendages of juvenile squid early in symbiosis.** Representative confocal microscopy images of AO-stained juvenile light organs, after exposure to no (APO), WT, or *ssrA*-deletion mutant (Δ*ssrA*) *V. fischeri*. Scale bar = 60 μm for all images. AO, acridine orange; APO, aposymbiotic; WT, wild type. (TIFF)

**S6 Fig. Effects of colonization by Δ*ssrA* on host physiology and health.** Visualization of SsrA transcript (magenta) in a whole-mount light organ, 24 h after colonization with a GFP-labeled Δ*smpB* strain of *V. fischeri* (green). A representative confocal image indicates that symbiont SsrA transcript is within the crypt epithelial cells. Scale bar = 10 **μ**m. (A) Visualization by HCR of SsrA transcript (magenta) in crypt #1 of a whole-mount light organ, 24 h after colonization with a GFP-labeled Δ*smpB* strain of *V. fischeri* (green), including orthogonal views of a confocal microscopy Z-stack; host nuclei (TO-PRO-3, blue). (B) Kaplan-Meier survival plots of juvenile squid colonized by WT, Δ*ssrA*, its genetic complement (Δ*ssrA* + *ssrA*), or the Δ*smpB* strain. Data are from replicate #1 (left), #2 (middle), or #3 (right). Survival-curve analyses used the log-rank Mantel-Cox test, with Bonferroni multiple-testing adjustment for pairwise comparisons. *P* value = 0.016 (S3 Data). (C) Kaplan-Meier survival plots of juvenile squid that were either single-colonized by WT or Δ*ssrA* or co-colonized at a 1:1 inoculum ratio with both WT and Δ*ssrA* ($n$ = 60); note that the WT and co-colonized data are coincident and significantly different from Δ*ssrA*. Log-rank Mantel-Cox test, with Bonferroni multiple-testing adjustment for pairwise comparisons. *P* value = 0.016 (S7 Data). (D) Respiration rates of newly hatched squid ("Hatch," $n$ = 5) or of animals after 24 h, that were either maintained APO ($n$ = 12) or colonized by WT ($n$ = 12), Δ*ssrA* ($n$ = 11), or Δ*lux* ($n$ = 11) strains. No significant difference between treatments was noted (S7 Data). (E) Internal yolk-sac areas, 2 d post colonization with WT, Δ*ssrA*, its complement (Δ*ssrA* + *ssrA*), the dark-mutant (Δ*lux*) or Δ*smpB* strains. Analysis used Kruskal–Wallis ANOVA, followed by Dunn's multiple comparison test ($n$ = 17). Data are represented as the median, with 95% confidence interval. *P* value code: ****<0.0001; ***<0.0002; **<0.001; *<0.021. ns: nonsignificant for all figures (S7 Data). APO, aposymbiotic; GFP, green fluorescent protein; HCR, hybridization chain reaction; WT, wild type. (TIFF)

**S7 Fig. Down-regulation of laccase-3 in the crypt epithelium requires the presence of symbiont SsrA.** (A) Localization of the laccase-3 transcript (magenta) in whole-mount light organs, 24 h post colonization. Representative confocal images showing laccase-3 expression in the crypt epithelia of APO (uncolonized) and WT, Δ*ssrA* or Δ*smpB*-colonized light organs; merged mid-section of Z-stack, and 3D reconstruction of the stack (S7 Data). (A') Quantification of laccase-3 signal using relative fluorescence intensity of a Z-series image of the light organ. *P* values were calculated using a 1-way ANOVA with TMC. (B) Quantification of

laccase-3 presence by HCR fluorescence signal intensity from a Z-series of light organs ($n = 5$), 3 h after incubation with OMVs isolated from either WT or Δ*ssrA* cultures. Addition of symbiont OMVs by themselves does not significantly change the expression of laccase-3 in the crypt epithelium (S7 Data). APO, aposymbiotic; HCR, hybridization chain reaction; TMC, Tukey's multiple comparison test; WT, wild type.
(TIFF)

**S1 Table. Oligonucleotide information.**
(DOCX)

**S1 Raw image. Raw image of Coomassie-stained PAGE gel of OMV proteins in S1C Fig.**
OMV, outer membrane vesicle.
(PDF)

**S1 Data. OMV RNA-seq.** Sheet 1: Counts in OMV and hemolymph samples. Sheet 2: Numerical values for Fig 1B. Sheet 3: Differential-expression analysis (Fig 1C). OMV, outer membrane vesicle; RNA-seq, RNA sequencing.
(XLSX)

**S2 Data. Numerical values Fig 2.** Sheet 1: CFU per squid. Fig 2A. Sheet 2: Number of hemocytes trafficking into the light-organ appendages after 16 and 18 h post colonization. Fig 2B. Sheet 3: Number of hemocytes trafficking into the light-organ appendages after 3 h inoculation with WT or Δ*ssrA* OMVs. Fig 2C. Sheet 4: Number of apoptotic nuclei per appendage. Fig 2D. Sheet 5: RLU per CFU of symbionts either within the light organ, or within a homogenate of the light organ, of a 24-h juvenile. Fig 2E. CFU, colony-forming units; OMV, outer membrane vesicle; RLU, relative light units.
(XLSX)

**S3 Data. Light-organ RNA-seq.** Sheet 1: Cytoplasm area ($\mu m^2$) values after 24 or 48 h post colonization. Fig 3A'. Sheet 2: Differential-expression analysis. Fig 3B. Sheet 3: Relative fluorescence intensity of laccase-3 transcript. Fig 3C'. Sheet 4: Relative expression values of light-organ laccase-3 after 24 h post colonization. Fig 3D. RNA-seq, RNA sequencing.
(XLSX)

**S4 Data. Numerical values Fig 4.** Sheet 1: Survival proportions of juvenile squid colonized by either WT, Δ*ssrA*, Δ*ssrA* + *ssrA*, or Δ*smpB*. Fig 4B, S6B Fig. Sheet 2: Dry weight of juvenile squid immediately after hatching ("Hatch") or at 4 d post hatching when kept APO or colonized with WT, Δ*ssrA*, or a nonluminescent mutant (Δ*lux*) strain. Fig 4C. Sheet 3: Quantification of internal yolk-sac area of juvenile squid immediately after hatching ("Hatch") or at 2 d post hatching colonized with WT or Δ*ssrA*. Fig 4D. APO, aposymbiotic; WT, wild type.
(XLSX)

**S5 Data. Numerical values Fig 5.** Sheet 1: Relative expression values of C3. Fig 5B. Sheet 1: Relative expression values of RIG-I. Fig 5B. C3, complement protein 3; RIG-I, retinoic-acid inducible gene-I.
(XLSX)

**S6 Data. Numerical values S1 and S2 Figs.** Sheet 1: Relative expression values of *ssrA* from bacteria cells fraction or OMV fractions. Cells grown in three different media: a tryptone-based medium (LBS) or LBS with the addition of either glycerol (32.6 mM) or GlcNAc (10 mM). S1A Fig. Sheet 2: $OD_{600}$ values over 24 h of bacteria growth in tryptone-based medium (LBS). S1A Fig. Sheet 3: $OD_{600}$ values over 24 h of bacteria growth in minimum medium. S1A Fig. Sheet 4: Motility in soft agar of WT or Δ*ssrA* cells measured as the diameter of the outer

migration ring at 3 and 7 h post inoculation. S2B Fig. Sheet 5: Respiration rates of WT, Δ*ssrA*, Δ*ssrA* + *ssrA*, and a nonluminescent *lux*-deletion mutant (Δ*lux*) normalized to $OD_{600}$. Fig 2C. Sheet 6: RCI between WT and Δ*ssrA* in co-inoculated light organs after 24, 48, and 72 h. S2D Fig. Sheet 7: Relative expression values of *ssrA* and *smpB*. S2E Fig. GlcNAc, *N*-acetyl-glucos-amine; LBS, Luria-Bertani salt medium; OD600, optical density at 600 nm; OMV, outer membrane vesicle; RCI, relative competitive index; WT, wild type.
(XLSX)

**S7 Data. Numerical values S6 and S7 Figs.** Sheet 1: Survival proportion of juvenile squid that were either single-colonized by WT or Δ*ssrA*, or co-colonized at a 1:1 inoculum ratio with both WT and Δ*ssrA* (*n* = 60). S6C Fig. Sheet 2: Respiration rates of newly hatched squid ("Hatch"), or of animals after 24 h, that were either maintained APO or colonized by WT, Δ*ssrA*, or Δ*lux* strains. S6D Fig. Sheet 3: Internal yolk-sac area values, 2 d post colonization with WT, Δ*ssrA*, its complement (Δ*ssrA* + *ssrA*), the nonluminescent mutant (Δ*lux*), or Δ*smpB* strains. S6E Fig. Sheet 4: Quantification of laccase-3 signal by HCR using relative fluorescence intensity of a Z-series image of the light organ. S7A Fig. Sheet 5: Quantification of laccase-3 presence by HCR fluorescence signal intensity from a Z-series image of light organs, 3 h after incubation with WT or Δ*ssrA* OMVs. S7B Fig. APO, aposymbiotic; HCR, hybridization chain reaction; OMV, outer membrane vesicle; WT, wild type.
(XLSX)

## Acknowledgments

We thank members of the McFall-Ngai and Ruby labs for helpful discussions. We especially thank Fredrik Bäckhed for excellent advice and helpful discussions and Susan Gottesman for suggesting the *smpB* mutant studies. Gabriela Aguirre and Susannah Lawhorn contributed valuable technical help. At UHM, Tina Carvalho provided scanning electron microscopy training, and the Histology & Imaging Core Facility performed tissue sectioning.

## Author Contributions

**Conceptualization:** Silvia Moriano-Gutierrez, Margaret J. McFall-Ngai, Edward G. Ruby.

**Data curation:** Silvia Moriano-Gutierrez.

**Formal analysis:** Silvia Moriano-Gutierrez, Tara Essock-Burns, Edward G. Ruby.

**Funding acquisition:** Margaret J. McFall-Ngai, Edward G. Ruby.

**Investigation:** Silvia Moriano-Gutierrez, Clotilde Bongrand, Tara Essock-Burns, Leo Wu.

**Methodology:** Silvia Moriano-Gutierrez.

**Project administration:** Margaret J. McFall-Ngai.

**Resources:** Clotilde Bongrand, Margaret J. McFall-Ngai, Edward G. Ruby.

**Software:** Silvia Moriano-Gutierrez.

**Supervision:** Margaret J. McFall-Ngai, Edward G. Ruby.

**Validation:** Silvia Moriano-Gutierrez.

**Visualization:** Silvia Moriano-Gutierrez, Tara Essock-Burns.

**Writing – original draft:** Silvia Moriano-Gutierrez.

**Writing – review & editing:** Silvia Moriano-Gutierrez, Clotilde Bongrand, Tara Essock-Burns, Margaret J. McFall-Ngai, Edward G. Ruby.

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
