## [Editor Report · Decision Letter 0]

15 Jun 2020

Dear Dr Ruby, 

Thank you for submitting your manuscript entitled "The non-coding small RNA SsrA is secreted by Vibrio fischeri to modulate critical host responses" for consideration as a Research Article by PLOS Biology.

Your manuscript has now been evaluated by the PLOS Biology editorial staff, as well as by an academic editor with relevant expertise, and I'm writing to let you know that we would like to send your submission out for external peer review.

Please re-submit your manuscript within two working days, i.e. by Jun 17 2020 11:59PM.

Kind regards,

Roli Roberts

Senior Editor

PLOS Biology

---

## [Decision Letter · Decision Letter 1]

30 Jul 2020

Dear Dr Ruby,

Thank you very much for submitting your manuscript "The non-coding small RNA SsrA is secreted by Vibrio fischeri to modulate critical host responses" for consideration as a Research Article at PLOS Biology and please accept our apologies for the time it has taken us to contact you with a decision on your study. Your manuscript has been evaluated by the PLOS Biology editors, an Academic Editor with relevant expertise, and three independent reviewers, whose expertise and comments you can find at the end of this email. Reviewer 2, Shelley Payne, has identified herself.

As you will see, all of the reviewers are quite supportive of the work, although they do raise several issues that will need to be addressed before we can consider its publication. While many issues can be addressed by additions/modifications to the text and re-analysis of sequencing data, some points would take a more effort to address. Of these, we find the competition assay suggested by the first two reviewers to be the most relevant, which can also address one major point of the 3rd reviewer (a function of ssrA independent of smpB). We would also like to stress the following comment of the 3rd reviewer “The Conclusions section should be changed into a proper discussion, where the authors should more critically reflect upon the strengths and weaknesses of their present model”. This reflects initial thoughts of our Academic Editor when reading the paper and some of the other reviewers points.

In all, in light of the reviews, we will be happy to invite a thorough revision of the work that addresses the reviewers' comments, including a competition assay, if feasible. Your revised manuscript is likely to be sent for further evaluation by at least some of the reviewers before we can make a decision on publication.

We expect to receive your revised manuscript within 3 months. Please email us (plosbiology@plos.org) if you have any questions or concerns, or would like to request an extension. At this stage, your manuscript remains formally under active consideration at our journal; please notify us by email if you do not intend to submit a revision so that we may end consideration of the manuscript at PLOS Biology.

**IMPORTANT - SUBMITTING YOUR REVISION**

*Re-submission Checklist*

*Published Peer Review*

*PLOS Data Policy*

*Blot and Gel Data Policy*

Sincerely,

Roli Roberts

Senior Editor,

rroberts@plos.org,

PLOS Biology

REVIEWS:

Reviewer's Responses to Questions

Reviewer identities:

Reviewer #1: No

Reviewer #2: Yes: Shelley M. Payne

Reviewer #3: No

Reviewer expertise:

Reviewer #1: effects of Vibro spp host development and metabolism

Reviewer #2: genetics of Vibrio sRNAs and metabolism

Reviewer #3: bacterial non-coding RNA; RNA in bacteria-host communications 

Review Comments to authors:

Reviewer #1: In the manuscript entitled "The non-coding small RNA ssrA is secreted by Vibrio fischeri to modulate critical host responses," the authors show convincingly that the small RNA ssrA, a transfer messenger RNA with a well-established role in ribosome rescue, is secreted in membrane vesicles and taken up by squid light organ epithelial cells. This attenuates the host innate immune response, allowing V. fischeri persist in the squid light organ. This is a fascinating and important story that adds to our knowledge of the interaction of host epithelia with commensals. Some observations are underdeveloped but do not detract from the overall impact of the reported findings. The following are specific comments:

Major comments:

1) Line 39: Reference 27 to V. cholerae OMV is not appropriate here. This work shows that a sRNA increases V. cholerae OMV production due to a decrease in expression of the outer membrane protein OmpA. OmpA, itself, is a virulence factor. The increase in OmpA expression in a sRNA mutant is most likely responsible for increased pathogen virulence, and no evidence is presented that the sRNA is transferred to the host cytoplasm or that it impacts the host epithelium directly.

2) Line 95: Hemolymph RNAseq: The majority of bacterial RNA's found in host hemolymph were tRNA's and ribosomal RNA's. 

a) Is this a consequence of the stability of these RNA's? Is there any evidence of RNA fragments or degradation?

b) For sRNA's, did the reads include the entire sRNA or were all parts of the sRNA equally represented in the reads? 

c) Were there any non-V. fischeri bacterial RNA's detected in the hemolymph? In other words, are all these bacterial RNA's coming from the light organ, which is mono-colonized?

d) The authors state that the same RNA's were detected in OMV's regardless of what medium V. fisheri was grown in. Is ssrA more abundantly transcribed than the other sRNA's detected in OMV's and hemolymph or is it selectively packaged into OMVs?

3) Line 111: The statement here that the OMV's differ only in the absence of ssrA seems too strong given the evidence provided. For instance, I wonder if the size and number of OMV's is the same and whether this might also affect the phenotype.

4) Line 177: Are there other reasons in vivo light emission could be less in the ssrA mutant? For instance, are the lux genes just has highly expressed? Is high cell density not signaled or sensed in vivo?

5) Figure 3B: In this heat map, it looks like ssrA rescue is very incomplete. In other words, the host response to the rescued ssrA mutant looks more like the response to the mutant itself than that to the wild-type bacterium. Is it possible that overexpression of ssrA results in less packaging into OMV's? 

6) Line 281: Is it also possible that OMV's formed outside the squid are different in some way or that OMV's that lack ssrA function in a dominant negative way. Perhaps an in vivo competition experiment would address the question of whether proximity to the epithelium is necessary for rescue by ssrA+ OMV's.

7) Line 313-Hemocyte experiments: It is not clear from this discussion whether the experiments in hemocytes are relevant to what is observed in squid light organ epithelial cells. Shouldn't activation of an innate immune RNA sensor lead to a more robust immune response? I'm not convinced that these experiments strengthen the story.

Minor comments:

1) Line 92: "one such study…" This wording is awkward and gives the impression that there were many similarly performed RNAseq studies of squid hemolymph, but only one gave this result. I suspect this is not the case.

2) Line 145: Should this be Figure 1D rather than 1B?

3) Line 150: Should this be Figure 1E rather than 1C?

4) Line 157: This sentence is confusing. Which signals and secreted molecules are the authors referring to? Is ssrA a signal or a bacterial effector?

5) Line 170: I apologize for the nit-picking, but technically the bacterial production of MAMP's has not been measured even qualitatively. I would say, instead, that the host response to these MAMP's is not different in the mutant.

5) Line 301: Is the implication that the juveniles do not take in nutrients other than those in the yolk sac? For the non-experts, it would be helpful to state this directly.

6) Line 305: This sentence is unclear. What happened to a "significantly greater extent as confirmed by electron microscopy"? 

Reviewer #2: The authors have established the squid Euprymna scolopes and its symbiont Vibrio fischeri as an outstanding model for understanding host-microbe interactions. In particular, they have studied every stage of colonization and persistence from both the host and bacterium perspective. This has allowed a detailed description of the genetics, biochemistry and biology of this complex interaction. In the current manuscript, they show that a small ncRNA produced by the bacteria enters host cells and manipulates the immune response toward the vibrios, thus preserving the mutualistic interaction. Failure to produce and deliver the ncRNA results in altered innate immune signaling and reduced persistence of the bacteria, a response that is detrimental to both host and bacterium. The authors use a variety of techniques to show that the effect of the ncRNA is on the host, not the bacterium, and to establish its localization and effects on host cells. Overall, this study will be of broad interest, and it provides insights into the interaction of the host with a beneficial bacterium, an area that is understudied compared to host-pathogen interactions.

Specific questions and comments :

 1. SsrA is found in vesicles and there is evidence in previous studies that vesicles can deliver cargo to host cells, but is there direct evidence that SsrA enters the host cell through vesicles? Can the RNA also be secreted into the environment and taken up by host cells, or does it require the vesicle for delivery? 

2.If SsrA is acting directly on host cells and is not required by the bacteria to maintain colonization, the wild type strain should be able to rescue the ssrA mutant, i.e. if the squid are infected with a 1:1 mixture of the wild type and mutant, the wild type strain should not outcompete the mutant. I am not sure whether competition experiments can be done in this model, since the bacteria are flushed daily, but it would be interesting to see if the mutant bacteria could persist in the squid that is also exposed to the wild type. 

Reviewer #3: The manuscript 'The non-coding small RNA SsrA is secreted by Vibrio fischeri to modulate critical host responses' by Moriano-Gutierrez et al. studies the influence of the secreted sRNA SsrA of the squid symbiont V. fischeri on its host. The authors discover that, in absence of the ssrA gene, bacterial persistence and luminescence of the squid's light organ is much reduced. Strikingly, this phenomenon is not dependent on SmpB, the natural protein partner of the ssrA gene product, the tmRNA. Therefore, a defect in the bacterial trans-translation ribosome rescue system cannot explain the observed phenotypes. They can detect Vibrio tmRNA outside the bacteria, in host tissue, supporting their model that it is carried along with other stable transcripts into the host via outer membrane vesicles. Finally, the wild-type and the ssrA deletion strains are shown to elicit different host responses (regulated genes in the squid), with an interesting effect on genes related to immune function. 

The manuscript is well written and presents new insight into the functions of secreted RNAs, and the reported observations are truly novel and exciting. Naturally, one wishes to see more mechanistic detail, for example, a better understanding of how tmRNA may influence host gene expression, whether bacterial tmRNA can be recovered with RIG-I, whether particular regions of the RNA are responsible for the observed phenotypes and so forth. However, we appreciate that this is not an easy experimental system, and would argue that some of those mechanistic question may be addressed in a follow-up paper. 

Major points:

* Since the authors, at this point in time, cannot conclusively prove that SsrA is actively secreted and taken up by the host, strong statement such as "SsrA signaling" and related terms should be softened throughout the paper. True, they do show that tmRNA is present in OMVs and that it can be detected by FISH in host tissue. But this does not prove secretion, it might as well be bacterial cell lysis (which would equally explain the detection of 16S rRNA in host epithelium; Fig. S3). 

* Are the effects on the host specific to the tmRNA of Vibrio fischeri? The authors should at least attempt to complement the ssrA deletion strain with ssrA genes from evolutionary related or distant bacteria, so see whether this "signaling" is a trait that evolved in this specific symbiosis. 

* Several of the figures/tables lack appropriate labeling (the labels of Fig 1C are missing the log base and it is not stated what was compared to calculate the fold change; Fig 2 contains several cut-off error bars; Fig 5B is only showing half of the error bars; Figs 2, 3C', 5B, S1A, S1A', S2B, S5C, S6A', S6B should show all data points).

* Table 1 gives the number of reads for certain ncRNAs in OMVs and haemolymph. This is uninformative without knowledge of the total number of reads and should be changed to give normalized percentages instead.

* Further, some of the data important for the major conclusions seem to be over/misinterpreted. In lines 113-116, the indicated difference shown in figure 2A might be statistically significant, but the decrease shown is marginal at best. Thus, the statement on line 119-120 is premature at this point. Line 174-175 and the abstract state that the ssrA knockout mutant reduces luminescence by 90% but Figure 2E does not support this statement. Similarly, all data shown in Fig 5B show strong variance but is still used for important conclusions such as that CIKS expression is only affected by tmRNA-deficient OMVs, whereas wt OMVs had no effect. Given the data presented, this seems like an over interpretation and the authors should avoid this conclusion and rephrase line 358-366 and move it to the discussion. This also means that the model shown in Fig 5C, much of which is based on Fig 5B, should be re-evaluated. Finally, the numbers of replicates for the data shown in Fig 5B are missing.

* The authors exclusively focus on the function of tmRNA as a secreted molecule, given that the smpB knockout background shows similar behavior as the wt. Yet, it is important to at least discuss the possibility of tmRNA having additional functions in the bacteria that might explain the observations. Could the host effects stem from a tmRNA function as a regulatory RNA in the bacteria? 

* The Conclusions section should be changed into a proper discussion, where the authors should more critically reflect upon the strengths and weaknesses of their present model and discuss it in more detail with the available literature. On the latter, see recent work from the Cossart laboratory in Paris discovering putative RNA export and delivery of bacterial transcripts to RIG-I (PMID: 31761719, PMID: 31951583). 

Minor points:

* When referring to the RNA, please use tmRNA; ssrA is the gene name. 

* Supplementary file 1: For easy data access, VF_XXXX locus tags might be more useful to the community than the shown GeneIDs, which do not seem to be part of the indicated annotations

* Lines 36-38: reference 16 does not seem relevant as it focuses on DNA and plasmids found in OMVs, it does not concern the host response.

* Lines 95-97: The authors refer to tRNAs and ncRNAs as ORFs. Please correct this. Additionally, it is odd that not a single mRNA was detected in these OMVs. 

* Lines 108-110: reference 46 is only refers to OMVs and spread in mice. Please adjust statement or reference included in the indicated lines ("…as reported in other animals.").

* Fig S2D: Why does the complementation of ssrA lead to such an increase in smpB mRNA levels?

* Lines 124-126: please indicate the samples per group. 

* Line 113 may refer to Fig S1C.

* Line 145 should refer to Fig 1D (twice).

* Line 148: the mentioned data not shown data seem very interesting and could be included in the manuscript

* Line 150 should refer to Fig 1E.

* Line 219: "their induction suggested that the host treats the ΔssrA colonization as an undesired infection" This statement should be re-phrased, there is currently no proof for it.

* Line 306: "the yolk sac had significantly decreased not only in area, but also to a significantly greater extent". Please re-phrase for clarity.

---

## [Decision Letter · Decision Letter 2]

14 Sep 2020

Dear Dr Ruby,

Thank you for submitting your revised Research Article entitled "The non-coding small RNA SsrA is released by Vibrio fischeri and modulates critical host responses" for publication in PLOS Biology. I have now obtained advice from two of the original reviewers and have discussed their comments with the Academic Editor. 

We're delighted to let you know that we're now editorially satisfied with your manuscript. However before we can formally accept your paper and consider it "in press", we also need to ensure that your article conforms to our guidelines. A member of our team will be in touch shortly with a set of requests. As we can't proceed until these requirements are met, your swift response will help prevent delays to publication. Please also make sure to address the data and other policy-related requests noted at the end of this email.

- a cover letter that should detail your responses to any editorial requests, if applicable

*Copyediting*

*Published Peer Review History*

*Early Version*

Sincerely,

Roli Roberts

Senior Editor,

rroberts@plos.org,

PLOS Biology

DATA POLICY REQUIREMENTS:

Regardless of the method selected, please ensure that you provide the individual numerical values that underlie the summary data displayed in the following figure panels as they are essential for readers to assess your analysis and to reproduce it: Figs 1BC, 2ABCDE, 3A’BC’D, 4BCD’, 5B S1AB, S2ABCDE, S6BCDE, S7A’B. NOTE: the numerical data provided should include all replicates AND the way in which the plotted mean and errors were derived (it should not present only the mean/average values).

REVIEWERS' COMMENTS:

Reviewer #1:

The authors have addressed all the reviewer's comments in detail. This is a fascinating manuscript that, in my view, should be accepted for publication.

Reviewer #2:

[identifies herself as Shelley M. Payne]

 The revised version of the manuscript has addressed my comments and concerns.

---

## [Editor Report · Decision Letter 3]

22 Sep 2020

Dear Dr Ruby,

On behalf of my colleagues and the Academic Editor, Luis Teixeira, I am pleased to inform you that we will be delighted to publish your Research Article in PLOS Biology. 

Early Version

PRESS 

Kind regards,

Alice Musson

Publishing Editor, 

PLOS Biology

on behalf of

Roland Roberts,

Senior Editor

PLOS Biology